# Estimating societal benefits from Nordic catchments: An integrative approach using a final ecosystem services framework

**Bart Immerzeel**[1]*, **Jan E. Vermaat**[1], **Gunnhild Riise**[1], **Artti Juutinen**[2], **Martyn Futter**[3]

**1** Faculty of Environmental Sciences and Natural Resource Management, Norwegian University of Life Sciences, Ås, Norway, **2** Natural Resources Institute Finland, Oulu, Finland, **3** Department of Aquatic Sciences and Assessment, Swedish University of Agricultural Sciences, Uppsala, Sweden

* bart.immerzeel@nmbu.no

**Data Availability Statement:** All relevant data are within the paper and its Supporting Information files.

## Abstract

Nordic catchments provide a variety of ecosystem services, from harvestable goods to mitigation of climate change and recreational possibilities. Flows of supplied ecosystem services depend on a broad range of factors, including climate, hydrology, land management and human population density. The aims of this study were: 1) to quantify the total economic value (TEV) of consumed ecosystem services across Nordic catchments, 2) to explain variation in ecosystem service value using socio-geographic and natural factors as explanatory variables in multiple linear regression, and 3) to determine which societal groups benefit from these ecosystem services. Furthermore, we tested the scientific rigour of our framework based on the concept of final ecosystem services (FES). We used a spatially explicit, integrative framework for ecosystem services quantification to compile data on final ecosystem services provision from six catchments across Denmark, Finland, Norway and Sweden. Our estimates showed a broad variation in TEV and in the proportion contributed by separate services, with the highest TEV of €7,199 ± 4,561 ha$^{-1}$ y$^{-1}$ (mean ± standard deviation) in the Norwegian Orrevassdraget catchment, and the lowest TEV of €183 ± 517 ha$^{-1}$ y$^{-1}$ in the Finnish Simojoki catchment. The value of material services was dependent on both geographic factors and land management practices, while the value of immaterial services was strongly dependent on population density and the availability of water. Using spatial data on land use, forest productivity and population density in a GIS analysis showed where hotspots of ecosystem services supply are located, and where specific stakeholder groups benefit most. We show that our framework is applicable to a broad variety of data sources and across countries, making international comparative analyses possible.

## 1 Introduction

Society depends on ecosystems in a multitude of ways: these can be easily visible and quantifiable processes as well as more concealed ones. The focus in land management decisions has historically been on maximising the production of material goods such as agricultural products

**Funding:** B.I. - 82263 - NordForsk - https://www.nordforsk.org/projects/integrating-nexus-land-and-water-management-sustainable-nordic-bioeconomy-biowater J.V. - 82263 - NordForsk - https://www.nordforsk.org/projects/integrating-nexus-land-and-water-management-sustainable-nordic-bioeconomy-biowater G.R. - 82263 - NordForsk - https://www.nordforsk.org/projects/integrating-nexus-land-and-water-management-sustainable-nordic-bioeconomy-biowater A.J. - 82263 - NordForsk - https://www.nordforsk.org/projects/integrating-nexus-land-and-water-management-sustainable-nordic-bioeconomy-biowater M.F. - 82263 - NordForsk - https://www.nordforsk.org/projects/integrating-nexus-land-and-water-management-sustainable-nordic-bioeconomy-biowater The funders had no role in study design, data collection and analysis, decision to publish, or preparation of the manuscript.

**Competing interests:** The authors have declared that no competing interests exist.

and forestry goods [1–3]. This can lead to societally sub-optimal results, since more concealed benefits received from ecosystems can be negatively impacted by a focus on marketable goods [1]. Researchers use ecosystem services frameworks to assess all the benefits society receives from its interaction with ecosystems, and to avoid the limited focus on those already quantified in standard economic analysis, like agricultural production [4]. To approximate a societal optimum in the benefits received from ecosystems, a complete overview of the net value of all relevant ecosystem services is useful. For instance, a landscape can create value both through agricultural production, as well as through recreational possibilities. The former can be quantified with established methods using statistics on agricultural production, the latter is typically not included in standard economic analysis. To reach a societal optimum, optimisation has to take both into account. Additionally, a framework that accounts for the manner in which underlying landscape and ecosystem characteristics influence these values will allow for an analysis of the effects of focused policy measures for different beneficiary groups in society, and can show possible synergies and trade-offs between ecosystem services.

However, applying the concept of ecosystem services raises questions: how do we actually benefit from ecosystems? Which processes are of value to us, and how do these benefits flow into societies [5]? Why are some services often ignored in policy decisions while others are not [3]? The open-ended nature of these questions has led to ongoing debate among researchers [6], including calls for 'a clear and robust definition' of what an ecosystem service is [7]. Currently, a variety of frameworks is available with definitions of variable precision [7–10]. Such heterogeneity hinders comparability across different studies, countries and likely also spatial scales [11,12], and it prevents well-informed generalisations for decision making and policy implementation [13].

In an effort to standardise ecosystem services accounts, the concept of final ecosystem services (FES) was introduced by Boyd and Banzhaf [8], and further elaborated on by Wallace [5]. Their definition of FES is: 'components of nature, directly enjoyed, consumed, or used to yield human well-being', in which the key term in our view is 'directly'. A FES requires a direct link to a component of nature and a human beneficiary, which differentiates this definition from other frameworks in general use, such as the Common International Classification of Ecosystem Services (CICES) [9] and Nature's Contributions to People by the Intergovernmental Science-Policy Platform on Biodiversity and Ecosystem Services [14], where quantified services can include outputs from other processes which would be intermediate steps under the FES definition. The concept of FES was also specifically designed to allow monetary valuation of ecosystem services to support environmental accounting [8]. This means that hard to quantify ecosystem services, such as heritage landscape value, are easily excluded from FES-based frameworks [7]. We, however, choose to use it because of its methodological rigour in definitions and its ability to quantitively compare effects for different groups of society and across different spatial scales.

Whereas Boyd and Banzhaf [8] argue for 'final' services as the ultimate flows that are truly used by society (see also Bateman, Brouwer [15]), Boerema, Rebelo [12] present six recommendations for ecosystem service assessments. In the current study we developed a framework that combines the rigor of the FES approach with locally available land use and statistical data and closely follows the recommendations from Boerema, Rebelo [12]. Its conceptual basis in literature, structure and content is presented in the methods section below. We use estimated monetary value as an indicator of the importance of the societal benefit of each service. We do so for comparative reasons, as monetary value has strong communicative power [16].

We developed this framework because of our interest in the importance of those ecosystem services that are overseen in spatial planning decisions and that may favour particular user groups or sectors [17], and to supply decision makers with enough information to take these

services into account. Additionally, it may stimulate further scientific debate on the definition of ecosystem services and the usefulness of ecosystem services frameworks. We populate our framework with empirical, locally and publicly available, statistical data from six catchments across four Nordic countries (Denmark, Finland, Norway and Sweden). These six catchments cover a considerable range in land use intensity, land use type, landscape and climate, and allow us a comparative approach beyond the scope of a single case study area (as in Queiroz, Meacham [18] or Zhou, Vermaat [19]), whilst we maintain the rigor of one common methodology, and apply sufficient resolution to analyse underlying explanatory factors and spatial relationships. Our interest is in the relative importance of different ecosystem services and we ask ourselves the following questions:

1. Which services are most important in these six Nordic catchments, and what underlying environmental and societal factors explain the variation in ecosystem services value?

2. Which stakeholder groups benefit from which services and do we observe potential spatial conflicts in their interests?

To structure our inquiry, we formulated the following hypotheses based on literature [18,20–22]:

1. Where primary sectors (forestry, agriculture) dominate land use, material services provide the most societal value;

2. Where population density is high, immaterial (e.g. recreational) services are of the highest value;

3. Recreational value is strongly linked to the presence of water.

Additionally, we aim to assess to what degree the framework meets the criteria for a scientifically rigorous method for ecosystem services quantification using criteria proposed by Boerema, Rebelo [12].

## 2 Method

### 2.1 Study site selection

We define our study sites using catchments, or river basins, since these form a naturally bounded system based on hydrology, which is a key factor in the supply of many ecosystem services [23] and are thus more suitable to ecosystem service estimation than administrative boundaries.

We selected our study sites to cover the variation in land use, population density and overall geography characterizing the Nordic countries. We sought at least one catchment in each of these countries (Table 1, Fig 1). We aimed to select a mix of catchments representing both agricultural, more densely populated areas and forested, less densely populated areas. When we selected multiple catchments per country, we did so based on maximal contrast in size, land covered by forest and agriculture, and population density. Haldenvassdraget, Vindelälven and Simojoki here represent sparsely populated, forest-dominated catchments in different geographic regions. Odense, Orrevassdraget and Sävjaån represent more densely populated catchments dominated by agriculture and urban areas.

### 2.2 Defining final ecosystem services

Our framework of FES builds on the 'Mononen-cascade' as applied in Boerema, Rebelo [12] and Vermaat, Immerzeel [21], and is based on the cascade perspective as described in

**Table 1. Study site descriptions showing size and land use for forest, agriculture, water bodies, urban area and nature reserves as percentage of the total area, as well as average population density and the proximity of the closest city to the catchment.**

|  | Halden-vassdraget | Orre-vassdraget | Odense | Simojoki | Sävjaån | Vindelälven |
|---|---|---|---|---|---|---|
| Country | Norway | Norway | Denmark | Finland | Sweden | Sweden |
| Catchment size (km$^2$) | 1,006 | 102 | 1,199 | 1,178 | 733 | 778 |
| Forested area (%) | 67 | 3 | 6 | 76 | 60 | 75 |
| Agricultural area (%) | 17 | 70 | 80 | 2 | 32 | 6 |
| Water area (%) | 6 | 15 | 1 | 1 | 1 | 2 |
| Urban area (%) | 1 | 8 | 12 | 0 | 2 | 1 |
| Nature reserve area (%) | 3 | 10 | 0 | 14 | 2 | 1 |
| Population per km$^2$ | 16 | 167 | 205 | 1 | 41 | 5 |
| Closest city (with distance from catchment in km) | Oslo (20) | Stavanger (15) | Odense (0) | Oulu (70) | Uppsala (0) | Umeå (20) |

We took land use values for forest, agriculture, water bodies and urban area from 2016 CORINE land cover data [24]. We took the area of nature reserve from GIS-databases of the national environmental agencies. We used population data from 2019 estimates by WorldPop (worldpop.org). We defined cities as having more than 50,000 inhabitants.

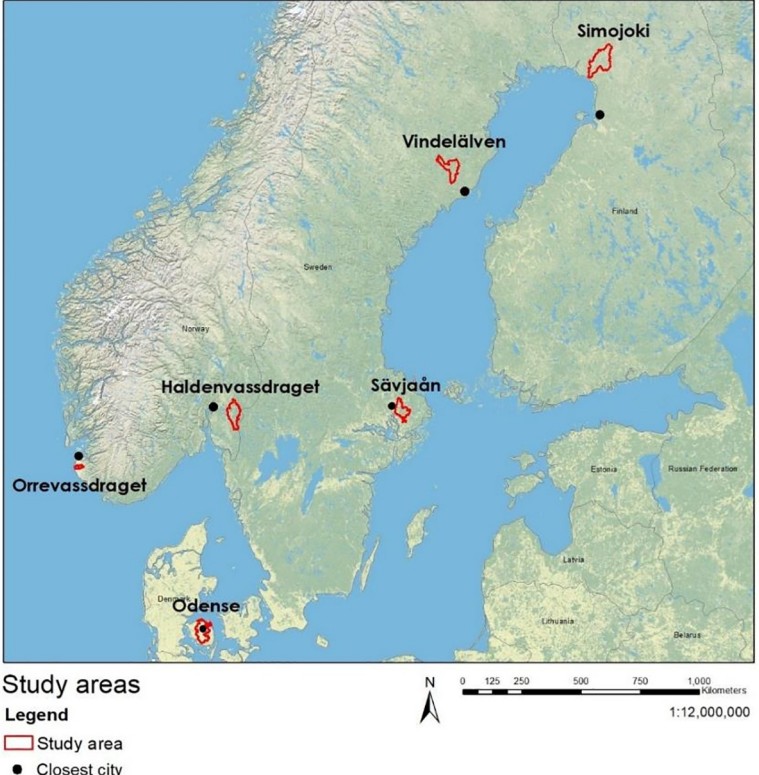

**Fig 1. A map showing the relative positions of the different study sites across the Nordic countries.** The basemap is provided by ESRI[a]. Study site boundaries are shown in red. Black dots show the city closest to the catchment as described in Table 1. This map illustrates the spatial range of study sites across the Nordic countries, as well as the range of dominant land use types. Orrevassdraget, Odense and Sävjaån are close to cities and in areas with relatively large areas of agricultural land, while Haldenvassdraget, Vindelälven and Simojoki are further from densely populated areas and contain relatively little agricultural land. [a] Esri. "World Topo Base". February 5, 2020. https://www.arcgis.com/home/item.html?id=3a75a3ee1d1040838f382cbefce99125. (September 14, 2020).

Mononen, Auvinen [25]. A key aspect of this framework is the stepwise flow from ecosystem structure and processes to the valuation of societal benefits. We define the FES as the point where, through a beneficiary, the ecosystem functioning flows into a societal benefit. We define FES based on the conditions set by Johnston and Russell [26], so that a biophysical outcome is a final ecosystem service if a beneficiary's welfare is influenced by it directly and with all other ecosystem outputs held constant, and an ecological process has served as an input to the biophysical outcome.

## 2.3 Quantifying FES consumption and TEV

We quantify consumption of FES using the concept of total economic value (TEV). The method of TEV quantification has been described in Pearce and Turner [27], TEEB [28] and de Groot, Alkemade [13], and has been used in a wide range of ecosystem services accounting studies including Costanza, dArge [29] and Wustemann, Meyerhoff [30]. This method monetizes the value of ecosystem services, including those services that are typically not included in standard economic analysis. By aggregating the values of ecosystem services into TEV in monetary terms, we allow for analysis of relative value of different services in a transparent way.

Using 1) the Mononen-cascade, 2) the CICES framework, 3) the above definition of FES and 4) established ecosystem services frameworks and applications in similar study sites [9,21,31,32], we constructed a list of FES consumed in our six study sites with their corresponding beneficiaries (Table 2). The list of beneficiaries aims to show every group in society directly benefitting from the ecosystem.

FES consumption is a flow over a given time period, where interaction with the ecosystem by a beneficiary yields a benefit. We chose to estimate the consumption of FES at an annual time step, allowing for the use of annual statistics as input data for quantification and valuation. This yields a TEV $yr^{-1}$ for each FES in each study site. Since we compare areas of varying size, we also estimated these flows in TEV $ha^{-1}$ $yr^{-1}$ for each FES. As the study sites also show large variation in population (Table 1), we also estimate TEV $inhabitant^{-1}$.

We base our quantification of ecosystem services on existing datasets, e.g. regional statistics on agricultural production and water extraction. This allows for assessment of actually consumed ecosystem services using multi-year means. An alternative method would have been modelling of the supply side processes that generate ecosystem services together with economic modelling of demand to generate estimated quantities of consumed ecosystem services. While modelling allows for more flexibility, we believe using real world statistics increases reliability and transparency. For an overview of all input data for each FES and how they link to final quantification, see S1.

We group FES into either material or immaterial ecosystem services, harkening back to the concept of ecosystem goods and services in Costanza, dArge [29] and Daily [4]. Material FES are tangible goods and energy, extracted from the ecosystem. Immaterial FES are intangible benefits, such as the enjoyment of recreating in nature, the prevention of flooding of property and the mitigation of climate change due to carbon sequestration. We choose this categorisation because it is easy to understand and is clearly linked to different beneficiaries as well as quantification and valuation methods (see Table 2).

For material FES, we quantify the mass of consumed goods. For immaterial FES, quantification is based on the amount of interaction: for recreation that is free to enjoy, we estimate the annual number of recreational trips based on survey data collected for Immerzeel et al. (in review), for carbon we quantify the amount sequestered annually, while for flood reduction we estimate the land area that is annually prevented from flooding due to water retention within the catchment [33].

**Table 2. List of final ecosystem services to quantify.**

| Type | Final ecosystem service | Beneficiary | What to quantify (per year) | Valuation method |
|---|---|---|---|---|
| Material | Supporting environment for crop production | Crop producers | Grains produced | Producer prices with ecosystem contribution coefficients |
| | | | Grass and fodder produced | Producer prices with ecosystem contribution coefficients |
| | | | Other crops produced | Producer prices with ecosystem contribution coefficients |
| Material | Supporting environment for forestry | Foresters | Roundwood removed | Producer prices with ecosystem contribution coefficients |
| Material | Availability of game | Hunters | Hunted big game | Producer prices |
| | | | Hunted small game | Producer prices |
| Material | Availability of peat | Peat extractors | Peat extracted | Producer prices with ecosystem contribution coefficients |
| Material | Potential for hydropower generation | Electricity generators | Electricity generated | Producer prices |
| Material | Availability of berries and mushrooms | Foragers | Berries gathered | Producer prices |
| | | | Mushrooms gathered | Producer prices |
| Material | Availability of water for drinking and processing | Water extractors | Water extracted | Producer prices |
| Immaterial | Recreational possibilities | Recreating visitors | Hunting licenses sold | License prices |
| | | | Fishing licenses sold | License prices |
| | | | Days of inhabitant recreation | Travel cost (Juutinen et al., in prep) |
| | | | Days of national visitor recreation | Travel cost |
| | | | Days of international visitor recreation | Travel cost |
| Immaterial | Mitigated climate change | Global society | Carbon sequestered in biomass | Social cost of carbon [36] |
| | | | Carbon sequestered in lake beds | Social cost of carbon |
| Immaterial | Prevented flood damage | Downstream property owners | Downstream area prevented from flooding | Land values and damage curves [37] |

This table shows for each ecosystem service whether it is a material service (benefit in terms of physical material and energy) or an immaterial service (related to experience or wellbeing), what we quantified and how we valued these quantified services. For detailed quantification methods and sources, see S1 File.

The final step in quantifying TEV is to convert quantities of consumed FES into monetary value. We use a variety of sources for this, depending on the type of FES. For marketable goods, we base our valuation on mean prices from official statistics for the period 2013–2017 or its closest available equivalent. Here Bateman, Mace [34] recommend to extract from the price all non-ecosystem sources of value in the value chain to end up with the contribution of the ecosystem in the price. We use ecosystem contribution coefficients for this, as described in Vallecillo, La Notte [35]. For agriculture and forestry, we use coefficients per EU country, using Swedish values for Norway as a proxy. For peat production in Finland, we use the same coefficient as for forestry in Finland. For FES that are produced without further human input and only require harvesting, like berries, game and mushrooms, we argue that the price the harvester receives for the good is a close approximation of the economic value of the FES. For non-material FES we estimate recreational value by taking travel cost values as estimated in Juutinen et al (in prep). This is data taken from a survey performed in the same six study sites, where respondents were asked how far they travel from their home to where they recreate most often within the area, and what mode of travel they typically use. This data was used to estimate the number of trips taken, as well as the willingness-to-pay for a recreational trip. For climate change mitigation we use the social cost of carbon [36] and for flooding we use land values and damage curves as described in de Moel and Aerts [37].

The source data underlying our TEV quantification includes spatial datasets (S2). This consists of data on land cover, population density, slope, soil type, stream networks, road networks and biomass productivity. This means we can convert our estimates of TEV into a spatial dataset and link this to these underlying landscape attributes. This allows us to visually analyse how the consumption of ecosystem services varies spatially and how these landscape attributes affect the spatial distribution of TEV. For this purpose, we created a vector data file in ArcMap 10.6, containing square polygons of 1 ha for each of our study sites. Using a set of if-statements (S3) to link our landscape attributes as well as our FES consumption to each separate cell, we distributed our TEV estimates over the study site per hectare.

## 2.4 Explaining variation

By dividing our study sites into subcatchments based on Strahler stream order, we were able to create a dataset of 223 nested spatial units with values on average consumption of specific FES and TEV per hectare as well as average values for various landscape and socio-geographic characteristics. We used linear regression in R (R Stats Package) to estimate correlations within and among catchments between FES and subcatchment characteristics.

We set up four models, each with a separate dependent variable: TEV, value from the supporting environment for crop production, value from the supporting environment for forestry, and value from recreational opportunities. We ran a multiple linear regression model for each predictand, using five landscape and socio-geographic characteristics as explanatory variables: average percentage of clay soil, average slope, average landscape diversity (using the Shannon Diversity Index or SDI on land cover data), average population density and fraction of open water area as part of total land cover. We chose these variables because they cover a wide variety of landscape characteristics: geophysical characteristics (soil and slope), characteristics directly affected by land management (SDI), societal characteristics (population density) and hydrology (surface water).

## 2.5 Stakeholder and conflict analysis

To analyse which groups in society benefit in which study sites, we created four stakeholder types: visitors, landowners, large extractors and global society. We then grouped the different beneficiaries linked to FES into these types (Table 3). This allowed us to analyse how the benefits for each group vary among and within study sites, by defining the main stakeholder group for each hectare cell, i.e. the group benefitting most in monetary terms.

Because different stakeholder groups receive different benefits for different land use types, conflicts may arise when land use changes from one type to the other. To show how possible land use change might impact stakeholder groups, we implemented two basic scenarios in each study site: in the first, all forest within 500m distance of agriculture transforms to agriculture, as long as the soil is not bedrock or moraine. In the second, all agriculture within 500m of forest transforms to forest. For Simojoki, we created two additional scenarios along the same

**Table 3. Stakeholder groups.** This shows per stakeholder group which beneficiary it contains, including the FES connected to that beneficiary.

| Visitors | Landowners | Large extractors | Global society |
|---|---|---|---|
| Hunters Availability of game | Crop producers Supporting environment for crop production | Water extractors Availability of water | Global society Mitigated climate change |
| Foragers Availability of berries and mushrooms | Foresters Supporting environment for forestry | Electricity generators Potential for hydropower generation | |
| Recreating visitors Recreational possibilities | Downstream landowners Prevented flood damage | Peat extractors Availability of peat | |

principle, but used a shift between forest and peat production areas. We then analysed for each of these scenarios what the impact is on annual TEV for each specific stakeholder group, by switching the average value of the original land use with the average value of the new land use type close to the original land use.

## 2.6 Testing the framework

Boerema, Rebelo [12] suggest the following six criteria for a successful ecosystem services framework (paraphrased):

1. Understand and explain the difference between the supply and demand side of ecosystem services, and be explicit of what you quantify.

2. Take into account the relationships between ecosystem services.

3. Use clear and consistent definitions for ecosystem services.

4. Measure all components that need to be measured for ecosystem service quantification.

5. Use scientifically rigorous and practically applicable measures and indicators.

6. Use scientific rigour in quality control, such as transparency, validity and uncertainty.

In addition, Hanna, Tomscha [2] give recommendations on quantification of riverine ecosystem services, stressing the need for complete quantification of all relevant ecosystem services, their interactions, and their spatial and temporal extent.

Since the above list is mostly qualitative rather than quantitative, we cannot test our data along these criteria statistically. Rather, we choose to discuss our results using previous literature to assess to what extent we comply with the criteria, and where shortcomings might have arisen. To judge our results on criteria 6, we compared our estimates to previous work on similar quantifications, and we performed a sensitivity analysis, showing how our results vary when changing our most uncertain source data values. We applied the following parameter changes:

- A 50% increase of the contribution of the ecosystem to crop production, to compensate for possible underestimation of the ecosystem contribution.

- A 50% reduction of value for carbon, to compensate for possible overestimation of the societal value of mitigating climate change.

- A 50% reduction in travel cost value for recreation, to compensate for possible overestimation of the value of recreational trips.

# 3. Results

## 3.1 Total economic value

Results of TEV estimation show large variation among the six study sites (Fig 2). TEV per study site is highest in Odense, with a total net value of more than 100 million euros per year. The area with the lowest net value estimate is Simojoki, with a TEV of around 20 million euros per year. This is partly caused by low population density and limited agricultural production, but also by low biomass growth rates, limiting the value of carbon sequestration (Table 4). The distribution of value between material and immaterial ES also varies across study sites. In Orrevassdraget by far most of the value is derived from recreational benefits, caused by high population density and high frequency of recreation. In Simojoki by contrast, more than half

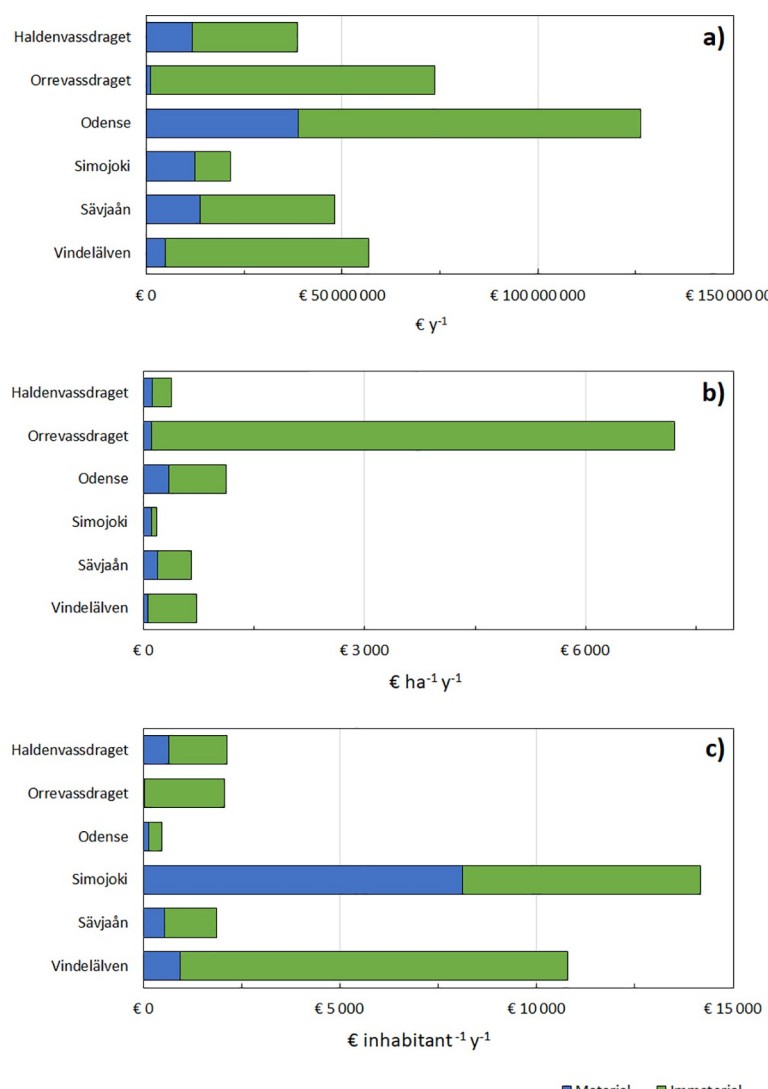

**Fig 2. Total economic value per study site, split out over material and immaterial ecosystem services.** a: The sum of all value consumed from ecosystem services per year in each study site. b: The same values, only divided by study site area in hectares. c: The same values, only divided by study site population.

of the net value is derived from material benefits. This is caused partly because of low population density, dampening the value of immaterial benefits, but also because of peat production, which generates some of the highest economic benefits per productive area (S1).

When standardising over area, a different picture appears. Orrevassdraget, the smallest catchment of the set but with high population density and a high share of land used for agriculture, yields by far the most value per hectare. This is mostly due to recreation enjoyed by a large number of inhabitants and visitors. Simojoki has the lowest areal net value, with some derived from mostly peat extraction, forestry and recreation along the river, but due to the very low population density and low carbon sequestration, there are few that benefit from the study site.

The negative effect of population is more strongly visible when looking at TEV per inhabitant of the area. Here the least densely populated areas stand out, signifying that there are decreasing marginal benefits over population. Simojoki and Vindelälven, mostly forested areas

**Table 4. Quantified ecosystem services, including corresponding CICES code [9] for reference, and their estimated monetary annual values in € ha$^{-1}$ year$^{-1}$ in each study site.**

| | Haldenvass-draget | Orrevass-draget | Odense | Simojoki | Sävjaån | Vindelälven |
|---|---|---|---|---|---|---|
| M–Agriculture (CICES 1.1.1.1/1.1.3) | 19 (58) | 99 (76) | 217 (333) | 11 (58) | 36 (62) | 6 (29) |
| Grains | 11 | 8 | 80 | 0.12 | 19 | 1 |
| Grazing and fodder | 7 | 87 | 99 | 11 | 13 | 4 |
| Other crops | 0.11 | 4 | 38 | 0.13 | 3 | 2 |
| M—Forestry (CICES 1.1.1.2) | 60 (49) | 9 (34) | 17 (465) | 28 (43) | 130 (107) | 50 (29) |
| Roundwood removal | 60 | 9 | 17 | 28 | 130 | 50 |
| M—Game (CICES 1.1.6.1) | 8 (3) | 0 (0) | 1 (1) | 0 (0) | 2 (1) | 1 (0) |
| Hunted big game | 8 | 0.22 | 1 | 0.21 | 2 | 1 |
| Hunted small game | 0.01 | 0.07 | 1 | 0.14 | 0.01 | 0.00 |
| M—Peat (CICES 1.1.5.2) | 0 (0) | 0 (0) | 0 (0) | 65 (513) | 0 (0) | 0 (0) |
| Milled peat | 0.00 | 0.00 | 0.00 | 65 | 0.00 | 0.00 |
| M—Hydropower (CICES 4.2.1.3) | 4 (0) | 0 (0) | 0 (0) | 0 (0) | 0 (0) | 0 (0) |
| Electricity generated | 4 | 0.00 | 0.00 | 0.00 | 0.00 | 0.00 |
| M—Foraging (CICES 1.1.5.1) | 0 (0) | 1 (3) | 0 (0) | 1 (0) | 1 (0) | 2 (1) |
| Berries gathered | 0.15 | 1 | 0.00 | 1 | 0.35 | 1 |
| Mushrooms gathered | 0.22 | 1 | 0.00 | 0.08 | 0.22 | 0.11 |
| M—Water consumption (CICES 4.2.1.1/4.2.2.1) | 26 (1) | 0 (0) | 109 (13) | 1 (0) | 17 (2) | 4 (0) |
| Water from catchment | 26 | 0.00 | 109 | 1 | 17 | 4 |
| I—Recreation (CICES 3.1.1) | 181 (171) | 7 080 (4 544) | 745 (1 748) | 31 (76) | 205 (210) | 512 (345) |
| Value of hunting | 1 | 1 | 5 | 0.06 | 0.23 | 0.13 |
| Value of fishing | 0.26 | 0.00 | 19 | 0.38 | 0.34 | 0.16 |
| Value of recreational trips–inhabitants | 162 | 5 775 | 677 | 21 | 168 | 373 |
| Value of recreational trips–national visitors | 17 | 1 099 | 39 | 9 | 34 | 133 |
| Value of recreational trips–international visitors | 1 | 205 | 6 | 0.00 | 3 | 5 |
| I—Carbon sequestration (CICES 2.2.6.1) | 85 (59) | 2 (4) | 16 (42) | 47 (29) | 260 (174) | 153 (99) |
| Carbon stored in biomass | 82 | 1 | 15 | 47 | 260 | 142 |
| Carbon stored in lakes | 3 | 1 | 1 | 0.33 | 1 | 11 |
| I—Flood prevention (CICES 2.2.1.3) | 1 (7) | 7 (52) | 15 (101) | 0 (1) | 4 (18) | 0 (2) |
| Water prevented from flooding land | 1 | 7 | 15 | 0.04 | 4 | 0.05 |
| **Total** | **384 (210)** | **7 199 (4 561)** | **1 121 (1 810)** | **183 (517)** | **656 (286)** | **728 (375)** |

Standard deviations are given in parentheses behind the main category mean. 'M' stands for material FES, 'I' stands for immaterial FES. Standard deviations are calculated using values per hectare cell. Note that these are means over total catchment area, not over area where the FES is consumed.

with low population density, shows the most value per inhabitant. This is due to the mix of services consumed: some agriculture, some forest with high consumption of timber, recreation of inhabitants and a relatively large number of visitors from outside the area (Table 4).

A spatial analysis of where the value is generated (Fig 3) shows that on first glance, landscape characteristics, especially river courses, along with population density, seem to have a strong impact on where value is generated. Looking at Haldenvassdraget, the central river valley contains the highest values per area. This corresponds to where agricultural land is located. A network of high value areas is also visible close to water edges, roads and high landscape diversity, since this is where people are most likely to recreate (Immerzeel et al., in prep). This effect is especially visible in Orrevassdraget, where the majority of value is derived from recreational trips. In Odense recreational value derived from the densely populated urban area can be seen radiating out from Odense city, with much of the value generated outside this core

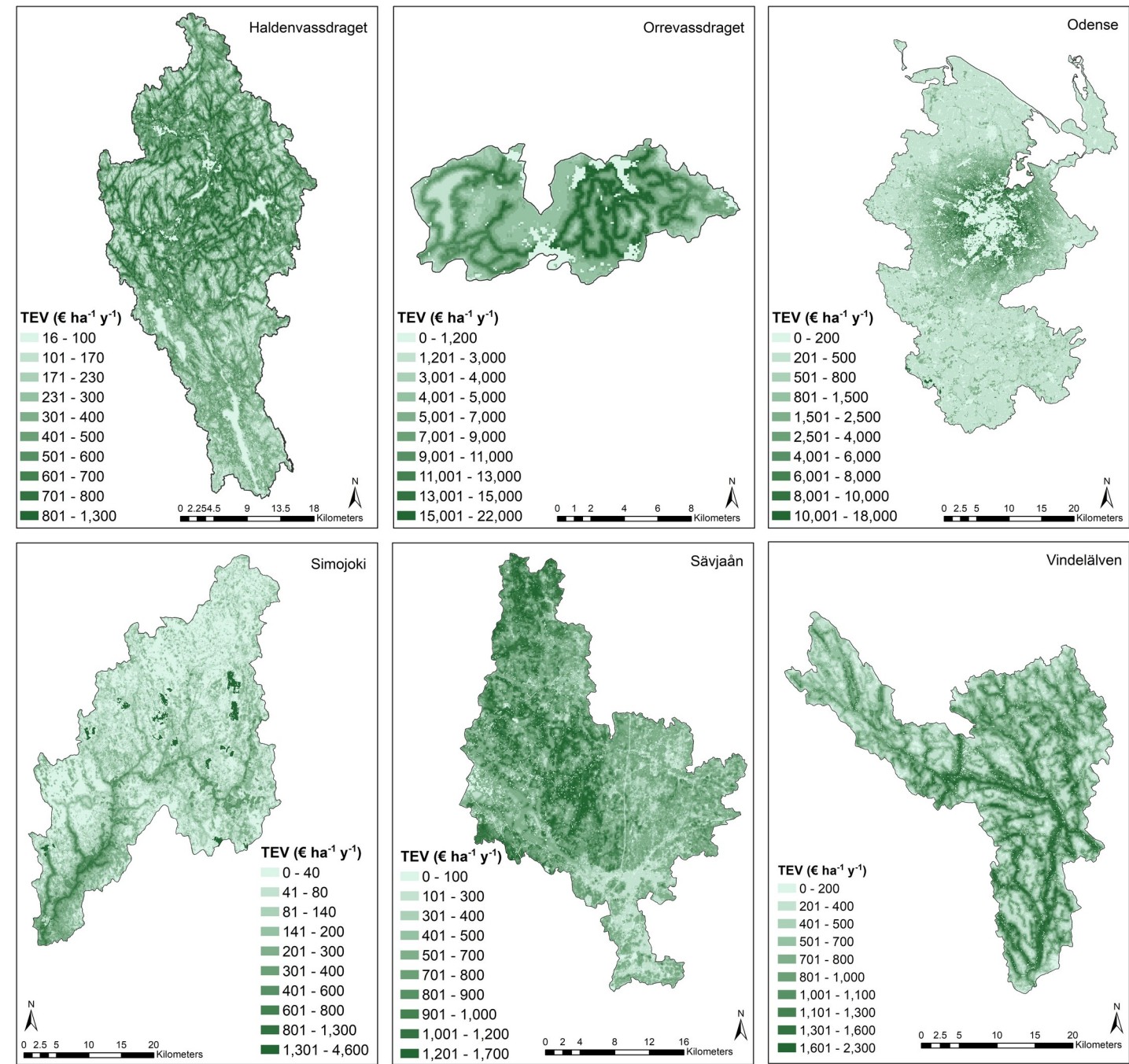

**Fig 3. Total economic value estimates per hectare per year for each study area.** Note the different colour scales. This reduces comparability among study areas, but increases the resolution of values shown within each study area.

coming from agriculture. This visible radius around the city is also caused by relatively short travel distances in this study site, due to a large proportion of relatively young city inhabitants going on shorter trips, often by bicycle (Immerzeel et al., in prep). In Simojoki two things stand out: the high value of the central river corridor, caused by agriculture along the stream and recreational salmon fishing, and the dark green areas of high value, where peat is extracted. In Sävjaån value is relatively uniformly spread, since forest productivity is high here,

**Table 5. Results of multiple linear regression models on subcatchment level.** Different FES values (top row) are dependent variables, and five study site characteristics (percentage clay soil, average terrain slope, average landscape diversity (SDI), average population in a 5 km radius around the cell and the fraction of water of total land cover in the subcatchment) are independent variables.

| | TEV | Agricultural value | Forestry value | Recreational value |
|---|---|---|---|---|
| | Coefficient (standard error) | | | |
| Intercept | 736.33 (216.45)*** | 77.35 (17.08)*** | 13.85 (13.00) | 615.30 (217.80) *** |
| Clay | -344.12 (293.32) | 131.04 (23.15)*** | -6.61 (17.62) | -474.67(295.16) |
| Slope | -4.41 (2.16)** | -1.11(0.17)*** | 0.08 (0.13) | -3.76 (2.18)* |
| SDI | - 323.58 (299.67) | -47.57(23.65)** | 60.30 (18.00)*** | -516.02 (301.54)* |
| Population | 384.32 (56.43)*** | 17.53 (4.45)*** | -7.06 (3.39)** | 372.61(56.78) *** |
| Water fraction | 2653.73(905.12)*** | -284.63(71.43)*** | 93.27 (54.38)* | 2626.63 (910.80)*** |
| N | 223 | 223 | 223 | 223 |
| Adj. $R^2$ | 0.22 | 0.36 | 0.08 | 0.20 |
| F-statistic | 13.36 | 26.23 | 5.08 | 12.09 |
| DoF | 217 | 217 | 217 | 217 |

***, **, * = = > Significance at 1%, 5%, 10% level respectively.

which increases value from both forestry and carbon sequestration. The western half of the area has a higher average value due to the proximity of Uppsala, increasing the recreational benefits received closer to the city. In Vindelälven the relative weight of recreational value is clearly visible in the contours of water edges and areas close to roads.

## 3.2 Explaining variation

Multiple linear regression using study site characteristics as explanatory variables for variation in the consumption of FES showed a number of significant correlations (Table 5). When looking at total economic value, we found that population density ($p<0.00$) and the fraction of surface water ($p<0.00$) show positive correlations, while the average terrain slope ($p<0.05$) shows a negative correlation. Zooming in on value from supporting environment for growing crops, availability of clay soils ($p<0.00$) and population density ($p<0.00$) show positive correlations to supplied value, while average slope ($p<0.00$), landscape diversity ($p = 0.05$) and the fraction of surface water ($p<0.00$) show negative relationships. Landscape diversity ($p<0.00$) and the fraction of surface water ($p = 0.09$) have a positive correlation to the value of the supporting environment for forestry, while population density ($p = 0.04$) has a negative correlation. For recreation, we found positive correlations with population density ($p<0.00$) and the fraction of water ($p<0.00$) in the subcatchments ($p<0.00$), and negative correlations with average slope ($p = 0.09$) and landscape diversity ($p = 0.09$). None of these models explain the majority of variance though, with a highest $R^2$ of 0.36 for value of the supporting environment for growing crops.

## 3.3 Stakeholder and conflict analysis

When taking the spatial variation of TEV consumption and zooming in on the distribution among the main stakeholder groups, some spatial patterns emerge, both within and among study sites (Fig 4). In Haldenvassdraget, Simojoki and Sävjaån, landscape characteristics are clearly visible: the main river valley with its fertile soil and low slope gradient appear as areas where landowners are the dominant stakeholder group. Additionally, in Simojoki, large areas along the main river have recreating visitors as the main stakeholder, since salmon fishing is one of the main recreational attractions in the area.

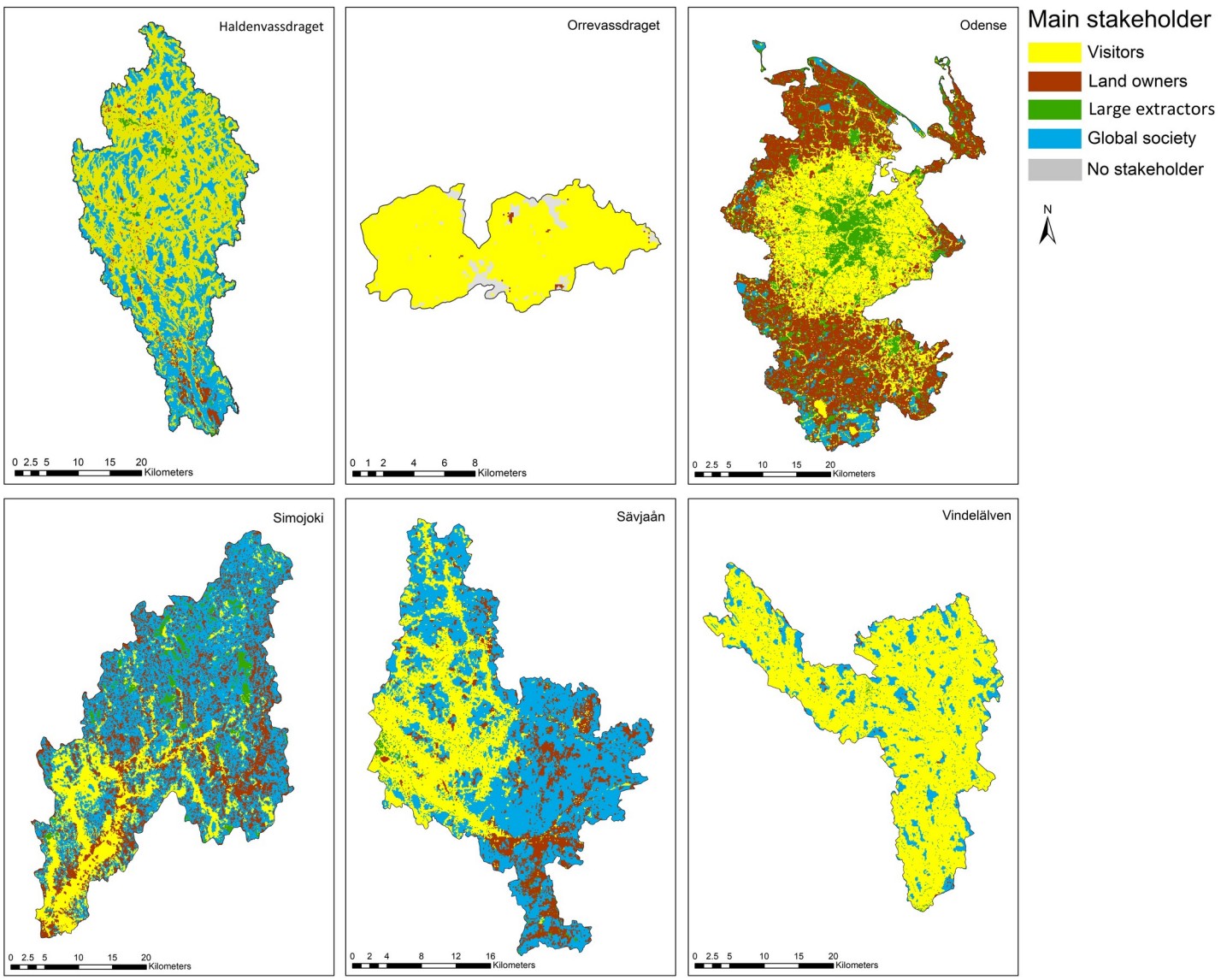

**Fig 4. Main stakeholder per hectare cell.** This shows per study area how FES consumption is spatially distributed by showing the stakeholder group with the highest TEV per cell.

Besides landscape characteristics, population density also shows clear effects on the spatial distribution of dominant stakeholders. In Odense, a clear radius can be seen around the city, illustrating that recreational visitors are dominant close to the city, while further away the weight shifts to landowners benefiting from the supporting environment for crop production. Large extractors dominate only where peat extraction occurs, or in urban areas where drinking water extraction is the main FES. Global society is the main stakeholder in the more remote areas in each study site. This also means that in densely populated areas like Orrevassdraget or Odense, global society is the main stakeholder in only a few small areas.

Moving on to possible conflicts, the effects of land use change on groups of stakeholders vary among study sites (Fig 5). When forested areas are transformed into agricultural land, the net effect varies among study sites. Global society loses everywhere, though in the northernmost sites, Simojoki and Vindelälven, this effect is negated by gains from other groups,

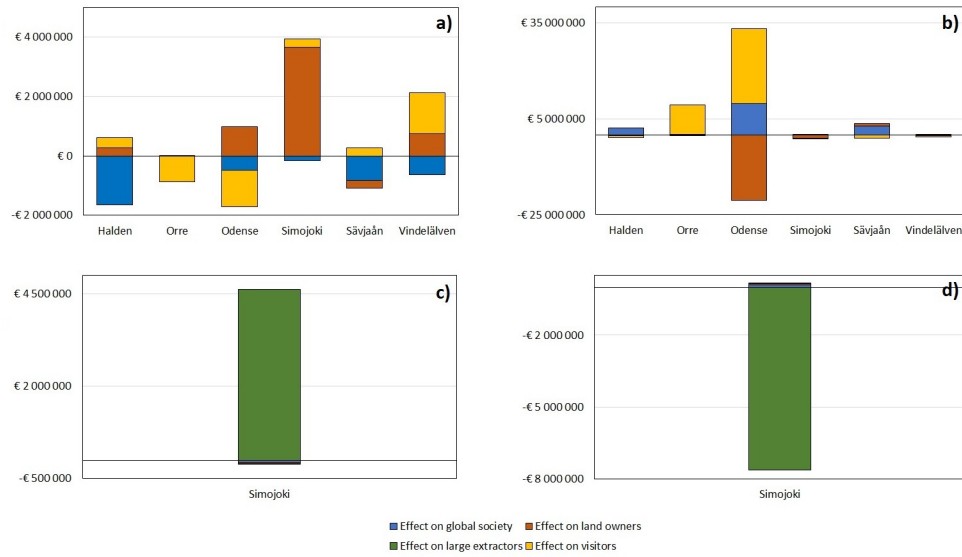

**Fig 5. Effects of land use change on stakeholder TEV.** a: shift from forest to agriculture. b: shift of agriculture to forest. c: shift from forest to peat extraction. d: shift from peat extraction to forest (see Methods section for details). For each of the scenarios, the total study site effect for each stakeholder group is presented as TEV in € ha$^{-1}$ year$^{-1}$.

since carbon sequestration is lower in these sites. Additionally, in areas already dominated by agriculture, the marginal value of forest for recreation is higher than in more mixed landscapes (Immerzeel et al., in prep), so there the value for visitors is also decreased. In Orrevassdraget and Odense this even leads to a net negative effect on TEV.

When moving in the opposite direction, converting agriculture to forest, the opposite effect on recreational value is apparent. Odense stands out here with a strong negative effect for land-owners, though this is offset by gains for both recreating visitors and global society. This is caused by the large number of scattered forest areas in the area: when all these expand, an area of around 73,000 hectares is transformed to forest, greatly reducing agricultural production in the catchment.

The results of this exercise also suggest that eliminating peat extraction in Simojoki will create a net loss to society of close to 8 million euros per year, due to the loss of value from extracted peat for large extractors. However, visitors, landowners and global society would all benefit.

## 3.4 Validity and partial sensitivity analyses

When comparing our results to previous studies with comparable study sites and methods, we find comparable results. Vermaat, Wagtendonk [38] use a similar method, estimating TEV in € ha$^{-1}$ year$^{-1}$ in six European river corridors before and after river restoration, based on market prices and stated preference studies. They estimate a mean TEV of €500 ha$^{-1}$ year$^{-1}$ before restoration for their Finnish site, mostly from crop production, and the southern Scandinavian sites closer to €1,000 ha$^{-1}$ year$^{-1}$, from a broader mix of FES. Remme, Edens [39] estimated value in € ha$^{-1}$ year$^{-1}$ for a province in the Netherlands for agriculture, drinking water, air quality regulation, carbon sequestration and recreation, finding a TEV of around €500 ha$^{-1}$ year$^{-1}$, mostly coming from agriculture and recreation. Lankia, Kopperoinen [40] estimated the value of recreation in various Finnish regions using survey data. The Simojoki area, for which we estimated a mean recreational value of €37 ha$^{-1}$ year$^{-1}$, is on the border of Lapland and

Northern Ostrobothnia, which Lankia, Kopperoinen [40] estimated to deliver recreational value of €15 and €58 ha$^{-1}$ year$^{-1}$ respectively. When looking at the highly productive agriculture in Odense, Lehmann, Smith [41] focused on the value of ecosystem services from agroforestry systems and found an average gross margin for agricultural production in Denmark of €1,067 ha$^{-1}$ year$^{-1}$, compared to our €217. They do not split out the ecosystem contribution to the gross margin, however. Our estimate for the ecosystem contribution is based on average producer prices of €930, close to their estimate. Nikodinoska, Paletto [42] estimated the value of various ES from the region around Uppsala, which also contains our Sävjaån area, using market prices, carbon permit prices and survey data. They found mean TEVs of around €1,200 ha$^{-1}$ year$^{-1}$ from forest areas and €600 from agricultural areas, compared to our combined average of around €650, based on an average producer price of around €750.

Sensitivity analyses on three of the more uncertain underlying variables show that doubling the value of the ecosystem contribution to crop production, and halving the value of carbon sequestration or recreational trips does not change the ranking of catchment TEV (Fig 6). However, effects of the changes vary among catchments. An increase of the contribution to crop production shows a particularly strong effect in Odense, increasing TEV by almost 20% due to its dependence on agriculture. Halving the value of carbon sequestration (presuming an overestimation of the societal value of mitigating climate change) mostly affects Sävjaån, with a reduction of 20% of TEV, due to the relatively high biomass productivity in that catchment. Reducing the value of recreational trips (presuming an overestimation of travel cost value) has a particularly strong effect in Orrevassdraget, with a reduction in TEV of 49%.

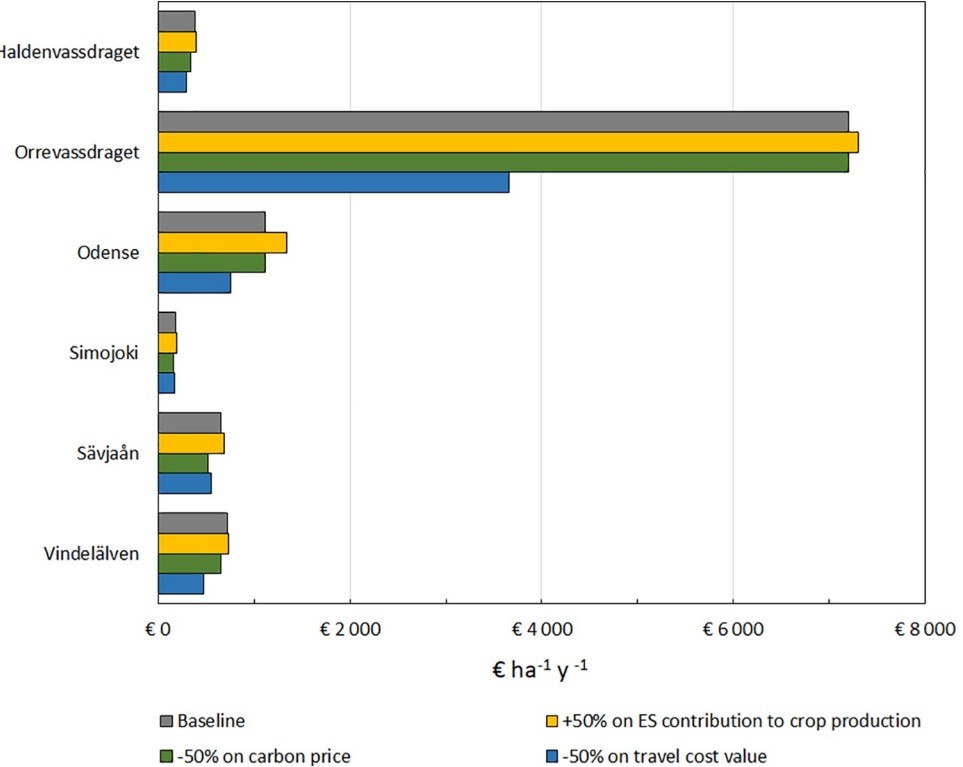

**Fig 6. Sensitivity analysis on three variables.** For each study site, TEV in € ha$^{-1}$ year$^{-1}$ is shown for the baseline, as well as for a change in three underlying variables.

## 4 Discussion

### 4.1 Interpretation of results

Our estimates show that catchments in Northern Europe provide society with value from a diverse source of ecosystem services. One of the main sources of value in our study sites is recreational value, being the main FES in Orrevassdraget, the area with the highest average value per hectare, as well as in Vindelälven, where that value is especially high due to the high travel cost per trip (Juutinen et al., in prep.). In all other study sites, it is also one of the FES with highest value. Since most of the recreation is enjoyed by local visitors from within the study site, one would assume a positive correlation between population density and value of recreation, and that is indeed what we found in our linear regression. The relationship between population density and ecosystem service value is in line with findings in other studies, for instance Vermaat, Wagtendonk [38] and Brander, Wagtendonk [43]. We also found a positive correlation between the fraction of surface water in an area and recreational value, supporting previous findings that water provides significant recreational value, for instance in Grizzetti, Liquete [44]. This also suggests possible conflicts in future land management change: agriculture, forestry and peat extraction can have a significant negative impact on water quality [45,46], and surveys in our study sites have shown that preference for recreation in these areas depends strongly on good water quality (Immerzeel et al., in prep.). Future studies on land use change should take these interactions into account. In Europe the Water Framework Directive, the EU directive that commits member states to achieve good qualitative and quantitative water state, also places demands on water management that need to be considered [47]. The negative relationship between terrain slope and recreational value likely has to do with access: roads and rivers, where recreation concentrates, tend to lie in relatively flat areas. Another major ecosystem service is the ability of the ecosystem to support agricultural production. Again, there is strong variation among our study sites, as well as within them. The availability of clay soil had a strong positive effect on value produced by agriculture. This reaffirms the notion that a combination of natural and societal characteristics affects TEV, emphasizing that value is created in the interaction. Besides soil and population, we found a significant negative effect of landscape diversity and slope on agricultural value. This is supported by previous work [48,49], showing that most agricultural value is created in relatively flat areas with uniform land use. For forestry we found few correlations between our selected study site characteristics and supplied value. This suggests that forestry is located where other land use is unprofitable or physically impossible, a 'leftover' land use [50].

The analysis of TEV and its variability additionally suggests that the effects of land management on the value of supplied ecosystem services might vary significantly, depending on the characteristics of the study site. This is also illustrated in our analysis of stakeholders and conflict. Changing land use in a similar manner can have a profoundly different effect on the benefits that society receives from ecosystems, depending on where this land use change takes place. For instance, landscape diversity appears to have a significant positive effect on recreational value. When changing from agriculture to forest, our analysis suggests that the effect on recreational value is positive in an agricultural area, but negative in an area that is already mostly forested. Alternatively, when transforming forest to agriculture, the benefits to land owners can be very high in an area where forest grows slowly, as in the northern Simojoki study site, but negative in an area like Sävjaån, where forest productivity is much higher and the value of FES consumed through forestry are higher than through agriculture.

These differences, along with large differences in share of benefit received by different stakeholder groups, indicate that conflict due to land use change is likely to occur when land management does not take them into account. Lee, Markowitz [51] have shown that

perception of local temperature change and understanding of anthropogenic effects on climate are strong predictors of concern for climate change, making it likely that public concern for climate change will increase over time. This can create conflict in study areas where global society would benefit from a transformation to forested area, whereas landowners would benefit from the opposite. Since our method only takes into account benefits from processes within the ecosystem, we do not include the effects of what happens with goods after extraction from the ecosystem. Therefore, we do not quantify carbon emissions from burning firewood or peat, or any change along the value chain towards consumers. However, with increasing public perception, pressure can increase for ceasing peat extraction altogether, and indeed, in Finland the public debate on what to do with this industry is increasingly volatile due to environmental concerns [52]. In the more densely populated areas, visitors and landowners also have competing interests, where visitors (and businesses dependent on them) appear to prefer more forest, while landowners are receiving benefits from maintaining the area for agricultural production.

### 4.2 Testing our hypotheses

Based on our findings, we partially dismiss the hypothesis that dominance of primary sectors in land use translates into dominance of FES delivered by these sectors. Forestry dominated catchments such as Haldenvassdraget and Vindelälven generate most of their value from the non-material FES recreation and carbon sequestration. In fact, material benefits only dominate in Simojoki, an area with very low population density. In areas with higher population density, such as Orrevassdraget, immaterial FES dominate, even in areas that are mostly covered by agricultural land use. This reinforces the notion of farmers serving as landscape stewards; the management of their lands serving not only their private interests, but a broader value of the landscape, as reflected in for instance the European Union's Common Agricultural Policy [53] and Norwegian subsidies to farmers for buffer strips along streams [54]. The spatial shift in dominance from material to immaterial services linked to population density can be most clearly seen in Odense (Fig 4).

This ties into the second hypothesis: where population density is low, immaterial FES tend to make a smaller contribution. Previous studies have found a positive relationship between population density and consumption of immaterial FES [38,55]. We also found that high population density in the vicinity of a cell increases recreational value (Table 4).

The third hypothesis, that recreational value is strongly linked to the availability of water, is clearly supported by our data. We found a significant positive relationship between the fraction of surface water in a subcatchment and the value of recreation generated in that subcatchment (Table 4).

### 4.3 Methodological limitations, uncertainty, and their implications

Using TEV has limitations, and there are alternative estimators of value. For example, TEV needs ecosystem services to be both quantifiable and of measurable value. Turner, Paavola [56] claim that TEV does not necessarily equate to a total system value, since it excludes the value of the system working as a whole, which they claim is more than the sum of its parts but cannot be measured in economic terms. There are also opponents of economic valuation of nature on principal, arguing that there is no objective measure of the value of an ecosystem, and that economic terminology is unsuitable for describing our relationship to nature [57]. While acknowledging these criticisms, we use TEV because it provides a transparent way of making quantitative estimates. This transparency gives unique power because it allows for comparative analysis, and monetary value works as a clear communicative method because it is easily understood and can be compared to costs and benefits of other societal activities.

The results of this study are based on a broad variety of sources. For quantification of FES consumption, we prioritised data with high detail and accuracy, which were national and regional statistics. This implies that we used separate sources for our study sites when crossing administrative boundaries, with different categorisations and data collection methods. When converting quantities to values, we again relied on a broad variety of sources to maximise precision (Table 2). Each of these however comes with its own range of uncertainty, and as Brander, Florax [58] and Schild, Vermaat [59] show, choice of valuation method can have a significant impact on value estimations. Therefore, compiling values based on different sources increases uncertainty in aggregate TEV estimates.

Another source of uncertainty is in the quantification method of marketed goods. The value of the FES is not in the value of the product, but in the part of that value generated by the ecosystem. We use ecosystem contribution coefficients from Vallecillo, La Notte [35]. However, these are country wide averages and not specified to our study sites. Additionally, Norway was not included in their study, nor was peat extraction, which we quantified by transferring their estimates from other values.

A third source of uncertainty is in the spatial analysis. Typically, spatial analyses of ecosystem service supply or demand use data from a single source for each variable. For land cover data for instance, CORINE data is often used, which allows for consistent comparison of land cover across European study sites [22]. However, these international datasets are typically generalised and of relatively low resolution. Due to the resolution we needed for our analyses on underlying drivers and stakeholders, we decided to use local datasets containing more detail and higher spatial resolution. This meant we compared outputs from datasets with different underlying methodologies and varying spatial resolution, which potentially increases uncertainty when comparing among different study sites. However, we argue this choice is worthwhile because it allows us the spatial resolution necessary to identify patterns, without claiming to know on a less than hectare level resolution what quantities of FES are supplied where.

In following the recommendations as given in Boerema, Rebelo [12], we argue we have succeeded or partially succeeded on all six recommendations (Table 6). When comparing our estimates to previous studies using similar methods in similar study sites, we find comparable results, which strengthens the argument that our estimates have sufficient validity according to recommendation six. However, since a full meta-analysis is beyond the scope of this study, we limited ourselves to five recent studies in North-western Europe. A basic sensitivity analysis on three of the underlying variables shows that sensitivity is low in general, but higher in study sites depending highly on a single FES, such as the large share of recreation in TEV in Odense. However, the general trends do not change, even when halving the strength of these variables.

## 4.4 Further research

One avenue of further research is to streamline the method we used for wider application. The current set-up is based on a broad variety of sources, making data collection labour intensive. A version based on international datasets, with a more user-friendly interface and quantification method, could possibly allow for large scale international comparisons that are relatively easy to implement and analyse.

A second direction for future research is to implement the dynamics of ecosystem processes in more detail. FES depend on a broad variety of ecological and environmental processes that are not currently included in our method, but are of importance because of interactions between human activity and these processes [60]. Nutrient retention and carbon cycling for instance affect multiple FES, but human activity conversely affects these processes as well [61].

**Table 6. Criteria for a framework of ecosystem services quantification based on Boerema, Rebelo [12], comments on the performance and success level in following the criteria for our framework.**

| Criterion | Comment | Success |
|---|---|---|
| Understand and explain the difference between the supply and demand side of ES, and be explicit of what you quantify. | We focused on the benefits society receives from interacting with ecosystems and we have explicitly tailored our framework to this, by incorporating a valuation step in monetary terms. By using FES, which focuses specifically on the point of interaction between ecosystem and beneficiary, we also choose to leave out the complexity of how ecosystem processes lead to a benefit. | Succeeded |
| Take into account the relationship between ES. | Since FES only quantify the value of the interaction between a specific stakeholder and the ecosystem, relationships between ES are minimised. Underlying processes within an ecosystem might impact several different FES and in that sense they are related, but because FES are focused on the process end point on the ecosystem side, we argue double counting is minimised: because every FES is explicitly linked to a stakeholder, there is no direct interaction. The only interaction is in changing the underlying base that supplies the FES, for instance when changing land use, and this we have taken into account in our stakeholder and conflict analysis. | Succeeded |
| Use clear and consistent definitions for ES. | In paragraph 2.2 we gave a definition of FES based on previous literature we believe to be clear and consistent, and that we adhered to throughout our data collection and analysis. | Succeeded |
| Measure all components that need to be measured for ES quantification. | It is likely that for some FES, such as the supporting environment for crop production, we measure more than the actual FES contribution. For other FES, such as climate change mitigation, we only take into account carbon sequestered into biomass and aquatic sediments, not those sequestered into soils. Since the majority of carbon fluxes is in biomass and not in soils [61], we argue this does not have a strong impact on our results. | Partially succeeded |
| Use scientifically rigorous and practically applicable measures and indicators. | Our measures and indicators are grounded in the FES definition and our quantification shows that they are practically applicable. Some of the indicators are proxies, but these are incidental, such as the contribution of the ecosystem to the producer price for extracted peat. | Partially succeeded |
| Use scientific rigour in quality control, such as transparency, validity and uncertainty. | We documented all sources and steps in the quantification process. These are available in S1. We compared our estimates to estimates from previous research using similar methodology on similar study sites, and found them to be within the same range of values previously reported. A basic sensitivity analysis showed how changes in value for our more uncertain inputs can affect the results. | Partially succeeded |

This affects immaterial value from recreation as well: Milner, van Beest [62] for instance show that the type of forest management impacts moose populations, which in the Nordic countries can have a strong effect on recreational value from hunting as well as the provision of game meat.

A third topic of further framework development is to apply scenario analysis. This builds upon the addition of interaction between ecosystem processes and FES consumption: quantified scenarios where for instance nutrient inputs from agricultural areas change, along with other environmental and societal variables can show the effects of land management change on the consumption of FES. This can also include more specific policy recommendations, showing which choices might affect stakeholders in different ways, as well as which instruments can be effective in reaching different stakeholder groups.

## 5 Concluding remarks

We have found that societal value from our six study sites is highly variable and derived from a variety of sources. Recreation, carbon sequestration and the supporting environment for agriculture tend to yield the largest benefit, though this is strongly dependent on study site characteristics. We have found a variety of environmental and socio-geographic characteristics covarying with FES value. Population density appears to be one of the key drivers for the most valuable FES, further strengthening the notion that it is in the direct interaction between people and nature that most value is created. Access to water is another key ecosystem characteristic driving FES value. We observe that different stakeholder groups value specific types of landscape differently, implying that land use change can lead to conflict. We show that global society, benefiting from climate change mitigation, can suffer if landowners or large extractors choose to increase their direct revenues by changing how they manage their land. Visitors aiming to enjoy the landscape can either suffer or benefit from a similar change, depending on the pre-existing state of the landscape.

We believe this application of the FES framework shows that a rigorous, consistent quantification of ecosystem services in varied landscapes across different countries is possible, and gives insight into what drives the variation in generated value. We believe it can be of value to decision makers by showing how different societal stakeholder groups benefit and may conflict, driving home the point that decision making should be tailored to local circumstances. Further research should focus on refining the toolset, a further integration of ecosystem processes underlying FES generation, and on scenario analysis for future land management, to aid in ensuring a sustainable and mutually beneficial relationship between society and nature.

## Supporting information

**S1 File. Truncated quantification spreadsheet.**
(XLSX)

**S2 File. Overview of spatial datasets.**
(DOCX)

**S3 File. Method description for spatial quantification.**
(DOCX)

## Acknowledgments

We would like to thank everyone who shared data and information with us, or who checked the quality of the data: Dennis Collentine at the Swedish University of Agricultural Sciences, Fatemeh Hashemi and Brian Kronvang at Aarhus University, Joy Bhattacharjee at the University of Oulu, Asta Harju and Tuija Vähäkuopus at the Geological Survey of Finland, Erkki Jokikokko at LUKE, Lars Selbekk at Marker municipality and Elin Kollerud at Utmarksforvaltningen. Maps in this paper were created using ArcGIS® software by Esri. ArcGIS® and

ArcMap™ are the intellectual property of Esri and are used herein under license. Copyright © Esri. All rights reserved.

## Author Contributions

**Conceptualization:** Bart Immerzeel, Jan E. Vermaat.

**Data curation:** Bart Immerzeel.

**Formal analysis:** Bart Immerzeel.

**Funding acquisition:** Jan E. Vermaat.

**Investigation:** Bart Immerzeel, Jan E. Vermaat, Gunnhild Riise, Artti Juutinen, Martyn Futter.

**Methodology:** Bart Immerzeel, Jan E. Vermaat, Gunnhild Riise, Artti Juutinen, Martyn Futter.

**Project administration:** Jan E. Vermaat.

**Supervision:** Jan E. Vermaat, Gunnhild Riise, Artti Juutinen.

**Validation:** Bart Immerzeel, Gunnhild Riise, Artti Juutinen, Martyn Futter.

**Visualization:** Bart Immerzeel.

**Writing – original draft:** Bart Immerzeel.

**Writing – review & editing:** Bart Immerzeel, Jan E. Vermaat, Gunnhild Riise, Artti Juutinen, Martyn Futter.

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
