## [Decision Letter · Decision Letter 0]

15 Apr 2021

PONE-D-21-04584

Estimating societal benefits from Nordic catchments: an integrative approach using a final ecosystem services framework

PLOS ONE

Dear Dr. Immerzeel,

Thank you for submitting your manuscript to PLOS ONE. After careful consideration, we feel that it has merit but does not fully meet PLOS ONE’s publication criteria as it currently stands. Therefore, we invite you to submit a revised version of the manuscript that addresses the points raised during the review process.

I think the submitted manuscript provides important strategy for assessing the ecosystem services. Both the reviewers have appreciated the study however suggested minor revisions. Moreover, I have also reviewed independently and agree with the reviewers comments. I, therefore, request the authors to revise the maunscript accordingly.

We look forward to receiving your revised manuscript.

Kind regards,

Lalit Kumar Sharma, Ph.D

Academic Editor

PLOS ONE

Journal Requirements:

Thank you for stating in the text of your manuscript that you obtained "signed informed consent for body donation for scientific investigation" from the patients before death. Please also add this information to your ethics statement in the online submission form.

Please provide the full name of the institution where cadavers were obtained.

We note that Figures 1, 3, 4 in your submission contain map images which may be copyrighted. All PLOS content is published under the Creative Commons Attribution License (CC BY 4.0), which means that the manuscript, images, and Supporting Information files will be freely available online, and any third party is permitted to access, download, copy, distribute, and use these materials in any way, even commercially, with proper attribution. For these reasons, we cannot publish previously copyrighted maps or satellite images created using proprietary data, such as Google software (Google Maps, Street View, and Earth). For more information, see our copyright guidelines: http://journals.plos.org/plosone/s/licenses-and-copyright.

4a, You may seek permission from the original copyright holder of Figures 1, 3, 4 to publish the content specifically under the CC BY 4.0 license. 

4b, If you are unable to obtain permission from the original copyright holder to publish these figures under the CC BY 4.0 license or if the copyright holder’s requirements are incompatible with the CC BY 4.0 license, please either i) remove the figure or ii) supply a replacement figure that complies with the CC BY 4.0 license. Please check copyright information on all replacement figures and update the figure caption with source information. If applicable, please specify in the figure caption text when a figure is similar but not identical to the original image and is therefore for illustrative purposes only.

Additional Editor Comments:

I think the submitted manuscript provides important strategy for assessing the ecosystem services. Both the reviewers have appreciated the study however I am also with an opinion that the authors must explain the basis of FES over the others. Moreover, I have also reviewed independently and agree with the reviewers comments. I, therefore, request the authors to revise the maunscript accordingly.

Reviewers' comments:

Reviewer's Responses to Questions

**Comments to the Author**

1. Is the manuscript technically sound, and do the data support the conclusions?

Reviewer #1: Yes

Reviewer #2: Partly

2. Has the statistical analysis been performed appropriately and rigorously? 

Reviewer #1: Yes

Reviewer #2: Yes

3. Have the authors made all data underlying the findings in their manuscript fully available?

Reviewer #1: Yes

Reviewer #2: Yes

4. Is the manuscript presented in an intelligible fashion and written in standard English?

Reviewer #1: Yes

Reviewer #2: Yes

5. Review Comments to the Author

Reviewer #1: 1. Summary of the research and your overall impression

The study presents a supply assessment and monetary quantification of ten ecosystem services in six catchments across four Nordic countries. The authors draw on publicly available statistical data on agricultural production and farm income, and land cover/use data and other spatial datasets. The authors apply published recommendations on how to conduct ecosystem services assessments, and provide an overview of how they meet these criteria. The authors conduct a sensitivity analysis for the ecosystem services more susceptible to suffer from data uncertainty. The authors also provide a scenario-building exercise to explore how conflicts between different stakeholders could emerge as a consequence of changes in land management.

I consider that this study provides an interesting assessment of the benefits provided to society by ecosystems depending on a range of biophysical and demographic factors, and should be interesting for, not only academic researchers, but likely also for decision makers and other stakeholders. The language is correct and clear, and the arguments flow nicely. Worth stressing is the well documented data sources provided in the supplementary materials, along with the data underlying the analysis.

Even though I consider the study is rather robust and deserves being considered for publication in the journal, below I provide some comments I hope will help the authors further increase the quality of the study.

2. Discussion of specific areas for improvement

1. Line 43: It is mentioned that ‘a complete overview of the net value of all relevant ecosystem services is useful’, but it seems that the authors don’t follow up too concretely on explaining ‘useful’ for what in the following sentence. Is there a way to be more concrete on this?

2. In the first paragraph of introduction (line 36-40) it is mentioned the historical land management focus on maximisation of production of material goods, and then use the TEV indicator to make comparable the ecosystem services assessed in the study. Even though the two doesn’t amount to the same, to me it looks slightly contradictory. Perhaps the authors could elaborate a bit on other type of values that could be used to assess the supply of benefits from ecosystems beyond monetary values? It might be appropriate providing such brief discussion point in subsection 4.3 on limitations.

3. Line 73: It is good to see that you developed the framework out of your interest for those overseen ecosystem services, but perhaps the authors could also point to some further reasons the audience why should appreciate your framework?

4. Line 84: ‘variation’ in what?

5. Lines 88-91: Is there a conflict between the first two hypotheses? What about places with both dominant primary sectors’ land uses and high population density (e.g. Odense, Orrevassdraget and Sävjaån)? I acknowledge that you touch upon this point in the discussion (lines 449-467), but perhaps worth to mention in the last part of introduction where you put forward your hypotheses that one doesn’t necessarily contradicts the other, if that is the case.

6. Lines 205-208: Since forest seems to a key land cover for the supply of ecosystem services in the study sites (most obviously for forestry, but also for game, foraging and recreation), I wonder why it was not included forest cover in the multiple linear regressions models. Perhaps it is because it is included in the SDI? If so, I assume then that water is also included there, but then is not clear why you also used fraction of water in the landscape as a separate variable. Perhaps it is interesting to explain just a bit more this point.

7. In relation to the previous comment, it might be interesting discussing briefly the criteria underpinning the selection of explanatory variables there.

8. I highly appreciate the very detailed methods’ section. However, I still can’t fully understand how you computed the monetary value of recreational based on travel costs. Since you refer to work not yet published for explanation (lines 181-182), perhaps a brief description of what travel costs include in your understanding might help the reader get an idea about the appropriateness of using such indicator for valuation of recreational services.

9. Line 301: ‘brackets’ could be substituted by ‘parentheses’.

10. Also in table 4, I wonder why the study sites are not given the full name as in the text and other figures. Space reasons?

11. Line 335: I think ‘...of dominant stakeholder’ should either be ‘the dominant stakeholder’ or ‘dominant stakeholders’.

12. Line 344: The reader might have forgotten by now which ones were ‘the northernmost sites’, i.e. Simojoki and Vildelälven I assume? Perhaps worth spelling their name here.

13. Line 356: From figure 5d is difficult to see why ‘visitors, landowners and global society would all benefit’. For landowners we see a little gain, but for visitors and global society? Is there anything missing there? Or I might have just missed something.

14. Line 410: It is the first the ‘Water Framework Directive’ is mentioned in the study. Is there a way the reader could get an idea what such framework is about, and why it is worth mentioning it here?

15. Line 416: I think something is missing in ’...emphasizing value is created in the interaction.’

Reviewer #2: I found this paper and its analysis to be clearly written and mostly well explained. The paper uses the FES ecosystem assessment method to explore spatial and other context differences in ecosystem supply and demand. My concerns with the manuscript as currently written are as follows:

The FES method is contrasted with the CICES and IPBESS approaches early on in the paper but I think the reader needs more explanation of the choice made to use FES. The FES approach is specifically designed to allow the monetary valuation of ecosystem services and as such does not have the coverage that other approaches claim to have. A number of ecosystem services are difficult if not impossible to value in monetary terms e.g. many culturally significant services such as heritage landscapes and other assets. These services arise in both high and low income economies I support the use of FES and also agree that monetised values carry weight in policymaking circles (especially finance ministries which control budgets). However, it is important to recognise its limitations, which the authors hint at in the latter sections of the paper. It would be clearer if these issues were dealt with at the start of the paper.

I don't agree that the key word in the FES approach is "direct" for me it is the word "final". The process by which individual welfare is changed need not literally be direct, particularly in non-use categories. Although the strength of the FES is that double counting can be avoided.

The paper combines the FES approach with the concept of TEV, but again it is important to clarify that TEV does not equal Total system value (TSV) ( see Turner et al 2003 in Ecological Economics for a discussion). In the paper the TEV is referenced by TEEB 2009 but it was first set out in Pearce and Turner 1991 Economics of Natural Resources and the Environment, Johns Hopkins Press Baltimore. I am highlighting the TEV and TSV distinction because the authors make mention of the term "societal optimum" on the first page of the paper. TEV is only a societal optimum in a restricted sense, because of the ESs that do impact on wellbeing but are not meaningfully expressed in monetary terms.

The authors use a range of monetary values based on different valuation methods, market price , travel cost and cost-based estimates and then introduce a sensitivity test. It was not clear to me how this resulted in the 50% increases and reductions in the chosen values. Why 50% ?. the text hints at a comparison with estimates in the published literature, as well as doubts/uncertaintity over the values chosen. Which was it, or was it both?. My concern with this part of the paper was highlighted by the remarks about the social cost of carbon. The reference given for this was Tol's paper. But this is a meta analysis of a number of estimates for SCC, so reducing it by 50% does not seem reasonable to me.

6. PLOS authors have the option to publish the peer review history of their article (what does this mean?). If published, this will include your full peer review and any attached files.

Reviewer #1: No

Reviewer #2: No

---

## [Author Response · Author response to Decision Letter 0]

7 May 2021

Reviewer #1: 1. Summary of the research and your overall impression

The study presents a supply assessment and monetary quantification of ten ecosystem services in six catchments across four Nordic countries. The authors draw on publicly available statistical data on agricultural production and farm income, and land cover/use data and other spatial datasets. The authors apply published recommendations on how to conduct ecosystem services assessments, and provide an overview of how they meet these criteria. The authors conduct a sensitivity analysis for the ecosystem services more susceptible to suffer from data uncertainty. The authors also provide a scenario-building exercise to explore how conflicts between different stakeholders could emerge as a consequence of changes in land management.

I consider that this study provides an interesting assessment of the benefits provided to society by ecosystems depending on a range of biophysical and demographic factors, and should be interesting for, not only academic researchers, but likely also for decision makers and other stakeholders. The language is correct and clear, and the arguments flow nicely. Worth stressing is the well documented data sources provided in the supplementary materials, along with the data underlying the analysis.

Even though I consider the study is rather robust and deserves being considered for publication in the journal, below I provide some comments I hope will help the authors further increase the quality of the study.

2. Discussion of specific areas for improvement

1. Line 43: It is mentioned that ‘a complete overview of the net value of all relevant ecosystem services is useful’, but it seems that the authors don’t follow up too concretely on explaining ‘useful’ for what in the following sentence. Is there a way to be more concrete on this?

We have included a further elaboration using a more concrete example (lines 44-47). 

2. In the first paragraph of introduction (line 36-40) it is mentioned the historical land management focus on maximisation of production of material goods, and then use the TEV indicator to make comparable the ecosystem services assessed in the study. Even though the two doesn’t amount to the same, to me it looks slightly contradictory. Perhaps the authors could elaborate a bit on other type of values that could be used to assess the supply of benefits from ecosystems beyond monetary values? It might be appropriate providing such brief discussion point in subsection 4.3 on limitations.

We mean to say that in standard economic analysis, the focus is on the monetary value of marketable, material goods, whilst in contrast, when using the TEV of ecosystem services, other services that are typically not monetized are now also included. We have added a sentence at the start of subsection 2.3 (methods, lines 146-149) to make this distinction more clear. In subsection 4.3 (discussion, lines 486-496). Here we have also added some discussion on other types of quantification and potential drawbacks of the TEV method.

3. Line 73: It is good to see that you developed the framework out of your interest for those overseen ecosystem services, but perhaps the authors could also point to some further reasons the audience why should appreciate your framework?

We have added further reasons for developing this framework (lines 82-86): to allow decision makers on land management to take all relevant ecosystem services into account, and to stimulate the scientific debate on the value of ecosystem services quantification.

4. Line 84: ‘variation’ in what?

In ecosystem services value. 

We have added this to the text (line 95).

5. Lines 88-91: Is there a conflict between the first two hypotheses? What about places with both dominant primary sectors’ land uses and high population density (e.g. Odense, Orrevassdraget and Sävjaån)? I acknowledge that you touch upon this point in the discussion (lines 449-467), but perhaps worth to mention in the last part of introduction where you put forward your hypotheses that one doesn’t necessarily contradicts the other, if that is the case.

There are indeed catchments with on average both high population density and land use dominated by primary production sectors, but when looking at more detailed spatial resolution (per subcatchment), typically areas either are densely populated or more intensively used for primary production. It is on this level that we test the hypotheses. 

6. Lines 205-208: Since forest seems to a key land cover for the supply of ecosystem services in the study sites (most obviously for forestry, but also for game, foraging and recreation), I wonder why it was not included forest cover in the multiple linear regressions models. Perhaps it is because it is included in the SDI? If so, I assume then that water is also included there, but then is not clear why you also used fraction of water in the landscape as a separate variable. Perhaps it is interesting to explain just a bit more this point.

We have found all land cover types to be key to the supply of certain ecosystem services. We did not include forest cover specifically because we thought it sufficient to use the diversity in the landscape (using SDI), as you indicate. We chose to include water separately (though it is indeed also included in SDI), because it cannot be altered by land use changes, such as shifting from agriculture to forest and vice versa, so stands separate from the other land cover types. It is also less directly related to provisioning services, like forest cover and its direct relationship to forestry, or cropland cover and its direct relationship to agricultural production. To clarify our choices of variable, we added another sentence to subsection 2.4 (lines 221-223).

7. In relation to the previous comment, it might be interesting discussing briefly the criteria underpinning the selection of explanatory variables there.

See the response at point 6.

8. I highly appreciate the very detailed methods’ section. However, I still can’t fully understand how you computed the monetary value of recreational based on travel costs. Since you refer to work not yet published for explanation (lines 181-182), perhaps a brief description of what travel costs include in your understanding might help the reader get an idea about the appropriateness of using such indicator for valuation of recreational services.

We have added extra sentences describing the type of data that was used and what was estimated with it (lines 191-194).

9. Line 301: ‘brackets’ could be substituted by ‘parentheses’.

Adjusted.

10. Also in table 4, I wonder why the study sites are not given the full name as in the text and other figures. Space reasons?

Indeed, as a way to save space. We have adjusted the names to their full name now while maintaining a one page table.

11. Line 335: I think ‘...of dominant stakeholder’ should either be ‘the dominant stakeholder’ or ‘dominant stakeholders’.

Adjusted to 'the dominant stakeholder group' for clarity.

12. Line 344: The reader might have forgotten by now which ones were ‘the northernmost sites’, i.e. Simojoki and Vildelälven I assume? Perhaps worth spelling their name here.

We referred to Simojoki and Vindelälven, yes. We have added their names for clarity.

13. Line 356: From figure 5d is difficult to see why ‘visitors, landowners and global society would all benefit’. For landowners we see a little gain, but for visitors and global society? Is there anything missing there? Or I might have just missed something.

The gains for visitors and global society are also there, only too small to be clearly visible in the current scale of the figure. We have edited the image to make it more clear.

14. Line 410: It is the first the ‘Water Framework Directive’ is mentioned in the study. Is there a way the reader could get an idea what such framework is about, and why it is worth mentioning it here?

We have elaborated this part of the text to give more context to the WFD (lines 424-426). It is worth mentioning here because it places demands on water quality that might be constraining land use change and land management (for instance when converting to more intensive agriculture, potentially leading to increased water pollution).

15. Line 416: I think something is missing in ’...emphasizing value is created in the interaction.’

Rewritten to 'emphasizing that value is created'.

Reviewer #2: I found this paper and its analysis to be clearly written and mostly well explained. The paper uses the FES ecosystem assessment method to explore spatial and other context differences in ecosystem supply and demand. My concerns with the manuscript as currently written are as follows:

The FES method is contrasted with the CICES and IPBESS approaches early on in the paper but I think the reader needs more explanation of the choice made to use FES. The FES approach is specifically designed to allow the monetary valuation of ecosystem services and as such does not have the coverage that other approaches claim to have. A number of ecosystem services are difficult if not impossible to value in monetary terms e.g. many culturally significant services such as heritage landscapes and other assets. These services arise in both high and low income economies I support the use of FES and also agree that monetised values carry weight in policymaking circles (especially finance ministries which control budgets). However, it is important to recognise its limitations, which the authors hint at in the latter sections of the paper. It would be clearer if these issues were dealt with at the start of the paper.

I don't agree that the key word in the FES approach is "direct" for me it is the word "final". The process by which individual welfare is changed need not literally be direct, particularly in non-use categories. Although the strength of the FES is that double counting can be avoided.

We agree that the choice for FES should have been expanded upon and we have now added this to the introduction. Our main motivations for choosing FES are due to its rigour and its possibilities for analysis, precisely because all included services are quantified using the same metric.

The paper combines the FES approach with the concept of TEV, but again it is important to clarify that TEV does not equal Total system value (TSV) ( see Turner et al 2003 in Ecological Economics for a discussion). In the paper the TEV is referenced by TEEB 2009 but it was first set out in Pearce and Turner 1991 Economics of Natural Resources and the Environment, Johns Hopkins Press Baltimore. I am highlighting the TEV and TSV distinction because the authors make mention of the term "societal optimum" on the first page of the paper. TEV is only a societal optimum in a restricted sense, because of the ESs that do impact on wellbeing but are not meaningfully expressed in monetary terms.

Indeed, the use of total economic value has been widely debated, for instance using the concept of total system value for a wider concept of value. We choose to use TEV to be able to make a quantitative comparative analysis. We have added a discussion under 4.3 on the appropriateness of using TEV (lines 486-496). In the methods section, under 2.3, we have also added a reference to Pearce and Turner (1991) (line 144).

The authors use a range of monetary values based on different valuation methods, market price , travel cost and cost-based estimates and then introduce a sensitivity test. It was not clear to me how this resulted in the 50% increases and reductions in the chosen values. Why 50% ?. the text hints at a comparison with estimates in the published literature, as well as doubts/uncertainty over the values chosen. Which was it, or was it both?. My concern with this part of the paper was highlighted by the remarks about the social cost of carbon. The reference given for this was Tol's paper. But this is a meta analysis of a number of estimates for SCC, so reducing it by 50% does not seem reasonable to me.

In 3.4, we describe two tests: firstly, we compared our estimates to estimates from previous results that is comparable (the same ecosystem services studied in northwestern Europe), as a check of validity. Secondly, as a partial sensitivity test, we manipulated underlying variables to see the effects on total economic value. We chose these variables for the following reasons: the value of the ecosystem contribution to crop production, because this variable combines two data sources; national/regional price statistics, and literature estimating the contribution of the ecosystem to crop value. Because it is based on two sources, we deemed it slightly more uncertain than other variables. The value of carbon sequestration, because it is based on mean values from a meta-analysis by Tol, whose dataset contained large variability and therefore we deemed it uncertain. The amount of recreational trips, because it is based on estimates from as of yet unpublished work. We have been involved in the collection of this survey data and stand by its quality, but since it has not been reviewed yet, we treat it as uncertain. The 50% changes were not based on a measure of uncertainty (since we cannot estimate such a measure with the data we have), but merely as an illustrative change to show the impact of a fairly extreme adjustment.

---

## [Editor Report · Decision Letter 1]

14 May 2021

Estimating societal benefits from Nordic catchments: an integrative approach using a final ecosystem services framework

PONE-D-21-04584R1

Dear Dr. Immerzeel,

We’re pleased to inform you that your manuscript has been judged scientifically suitable for publication and will be formally accepted for publication once it meets all outstanding technical requirements.

Kind regards,

Lalit Kumar Sharma, Ph.D

Academic Editor

PLOS ONE
---

## [Editor Report · Acceptance letter]

18 May 2021

PONE-D-21-04584R1 

Estimating societal benefits from Nordic catchments: an integrative approach using a final ecosystem services framework 

Dear Dr. Immerzeel:

I'm pleased to inform you that your manuscript has been deemed suitable for publication in PLOS ONE. Congratulations! Your manuscript is now with our production department. 

Kind regards, 

on behalf of

Dr. Lalit Kumar Sharma 

Academic Editor

PLOS ONE